# Unresectable Ovarian Cancer Requires a Structured Plan of Action: A Prospective Cohort Study

**DOI:** 10.3390/cancers15010072

**Published:** 2022-12-22

**Authors:** Gatske M. Nieuwenhuyzen-de Boer, Malika Kengsakul, Ingrid A. Boere, Helena C. van Doorn, Heleen J. van Beekhuizen

**Affiliations:** 1Department of Gynecologic Oncology, Erasmus MC Cancer Institute, University Medical Center Rotterdam, 3015 GD Rotterdam, The Netherlands; 2Department of Obstetrics and Gynecology, Albert Schweitzer Hospital, 3318 AT Dordrecht, The Netherlands; 3Department of Obstetrics and Gynecology, Panyananthaphikkhu Chonprathan Medical Center, Srinakharinwirot University, Nonthaburi 11120, Thailand; 4Department of Medical Oncology, Erasmus MC Cancer Institute, University Medical Center, 3015 GD Rotterdam, The Netherlands

**Keywords:** advanced-stage ovarian cancer, unresectable disease, cytoreductive surgery, treatment, survival

## Abstract

**Simple Summary:**

Patients with unresectable ovarian cancer during cytoreductive surgery for advanced-stage ovarian cancer are typically underreported. Hence, knowledge of further postoperative treatment and survival in case of unresectable disease during surgery is limited. The aim of this study is to address the knowledge gap about postoperative treatment and survival of patients whose surgery was abandoned due to unresectability after abdominal exploration. This is a post hoc analysis of the PlaComOv study, a randomized controlled trial. In this prospective study, 27 patients with the unresectable disease are described. Treatment was divers, ranging from the cessation of treatment to, predominantly, one or several lines of chemotherapy with or without maintenance treatment with bevacizumab and/or PARP inhibitors. The median overall survival after surgery was 16 (IQR 5–21) months (95%CI 14–18). At 24 months of follow-up, four patients (15%) were alive with the disease.

**Abstract:**

Background: Patients with unresectable disease during cytoreductive surgery (CRS) for advanced-stage ovarian cancer are underreported. Knowledge of treatment and survival after surgery is limited. The aim of this study is to address the knowledge gap about postoperative treatment and survival of patients whose surgery was abandoned due to unresectability after abdominal exploration. Methods: Women with FIGO stage IIIB-IV epithelial ovarian cancer whose disease was considered to be unresectable during surgery were included in this prospective study, a post hoc analysis of the PlaComOv study. The unresectable disease was defined as the inability to achieve at least suboptimal CRS without attempted CRS after careful inspection of the entire abdomen. Preoperative clinical data, perioperative findings, postoperative treatment and survival data were analyzed. Results: From 2018 to 2020, 27 patients were included in this analysis. Treatment ranged from the cessation of treatment to one or several lines of chemotherapy with or without maintenance therapy. The median overall survival was 16 (IQR 5–21) months (95%CI 14–18). At 24 months of follow-up, four patients (15%) were alive. Conclusions: This study indicated a two-year survival of 15%. Optimal treatment strategies in terms of survival benefits are still ill-defined. Further study of this specific group of patients is warranted. We advocate an (inter)national registry of patients with unresectable cancer and comprehensive follow-up.

## 1. Introduction

Despite extensive preoperative examinations, during abdominal exploration, it may be found that cytoreductive surgery (CRS) is impossible because of extensive disease in patients with advanced-stage epithelial ovarian cancer (AEOC). In those patients, surgery has to be abandoned because of unresectable disease. Patients with unresectable ovarian cancer are typically underreported or included in a suboptimal CRS (>1 cm residual tumor) group [1]. 

Ovarian cancer is the eighth most occurring cancer in women, with almost 314,000 new cases and more than 207,000 deaths worldwide within 2020 [2]. At present, the standard treatment of advanced-stage epithelial ovarian cancer (AEOC) consists of cytoreductive surgery (CRS) and platinum-based chemotherapy (mainly six courses, three weekly: neoadjuvant carboplatin (AUC6) with paclitaxel (175 mg/m^2^), followed by maintenance therapy with poly adenosine diphosphate-ribose polymerase (PARP) inhibitor in selected patients [3,4]. The timing of surgery may be as primary CRS or as interval CRS after three cycles of neoadjuvant chemotherapy [5]. Complete resection of all macroscopic diseases (at primary or interval surgery) is the strongest independent variable in predicting overall survival [6]. 

In the case of unresectable disease for patients with AEOC, knowledge of further postoperative treatment and survival is limited. 

To address this knowledge gap, we analyzed data from patients whose surgery was abandoned due to unresectability after abdominal exploration [7,8]. The aim of this study was to establish a detailed account of individual patients’ treatment along with a report on overall survival. 

## 2. Materials and Methods

### 2.1. Study Design and Patients

From 2018 to 2020, 327 patients with AEOC International Federation of Gynecology and Obstetrics (FIGO) stage IIIB-IV who were suitable for CRS and chemotherapy were eligible for inclusion in the PlaComOv study. The PlaComOv study was a multicenter randomized controlled trial to investigate the use of the PlasmaJet Surgical device during CRS (Figure 1) [8]. The study was approved by the Medical Ethics Review Board of the Erasmus University Medical Center Rotterdam, the Netherlands (NL62035.078.17). Details of the PlaComOv study and main study outcomes were published previously [7].

In this prospective cohort study, we included patients in whom the disease was considered unresectable during surgery. ‘Unresectable’ was defined as the inability to achieve at least suboptimal CRS (tumor lesions >1 cm) without attempted CRS after careful inspection of the entire abdomen and after intra-operative consultation with the anesthesiologist and the gynecologic oncologic surgeon.

At diagnosis, a laboratory test of the CA-125 level and a computerized tomography (CT) scan was performed, followed by discussion in a multidisciplinary tumor board meeting to determine whether primary CRS or neoadjuvant chemotherapy (NACT) followed by interval CRS was appropriate [5,9]. 

The NACT regimen consisted of three cycles of intravenous paclitaxel (175 mg per square meter of body-surface area) and carboplatin (area under the curve of 6 mg per milliliter per minute) with an interval of three weeks for each cycle [4,5]. A CT scan after three cycles of NACT was performed to evaluate the degree of tumor response. In the subsequent tumor board meeting, patients with (partial) response or at least stable disease were considered eligible and planned for interval CRS unless strict criteria for the unresectable disease were present [10]. 

All surgical procedures were performed by well-trained gynecological oncologists and by an oncological surgeon when indicated. Postoperatively, the possibility of continuing first-line chemotherapy was discussed with the patient. 

In the current study, preoperative clinical data, perioperative findings and postoperative treatment and survival data were analyzed. The pre-operative data which were analyzed were age, BMI, histology, FIGO stage, somatic mutation status of BRCA1 and BRCA2, level of Cancer antigen 125 at diagnosis and after NACT, WHO performance status, comorbidity and polypharmacy.

Normal CA-125 level was defined as <35 kU/L [11]. The reduction of CA-125 level after NAC was calculated.

Multimorbidity was defined as morbidity in three or more organ systems. Polypharmacy was defined as the use of five or more medicines for at least 90 days [12].

### 2.2. Statistical Analysis

Categorical variables were presented as numbers, and continuous variables were presented as mean ± standard deviation (SD) or median with interquartile range (IQR) as appropriate. Overall survival (OS) was calculated from date of surgery to time of death or last follow-up. The survival analysis was performed using Kaplan–Meier method. Statistical significance was considered when p-value *p* < 0.05. The analysis was performed using IBM SPSS statistics for Windows, version 22.0 (Armonk, NY, USA: IBM Corp).

## 3. Results

Of 327 patients in the PlaComOv study, 27 patients (8.3%) had the unresectable disease and were included in this analysis (Figure 1). The baseline characteristics are presented in Table 1. The mean age was 70 (SD 10.5, IQR 29–82) years. The mean body mass index was 25 (SD 6.4, IQR 17.2–47.9) kg/m^2^. Nine patients (35%) were classified as World Health Organization (WHO) performance status 0, 13 patients (50%) as WHO 1, and 4 patients (15%) as WHO 2. There were eight patients (30%) with multimorbidity (morbidity affecting three or more organ systems), and six patients (22%) had polypharmacy (using five or more medicines for at least 90 days at diagnosis). The median CA-125 at diagnosis was 660 kU/L (IQR 120–16,054). After three cycles of NACT, the median CA-125 was 81 kU/L (IQR 13–2695). The mean percentage of CA-125 reduction after NACT was 76% ± 25 (IQR 19.4–98.6). Seven patients (26%) had a drop of ≥95%.

All but one patient had high-grade serous adenocarcinomas; one patient had low-grade serous adenocarcinomas. Sixteen (59%) patients were FIGO stage III and nine (41%) were FIGO stage IV. The somatic mutation status of BRCA1 and BRCA2 was tested in twenty patients, and all were negative.

### 3.1. Surgical Findings

Table 2 reported the intraoperative findings and reasons for abandoning surgery. One patient was eligible for interval CRS after a diagnostic laparoscopy, but an optimal CRS was not feasible with laparotomy. All patients had ascites and extensive peritoneal carcinomatosis (defined as >200 tumor spots on the peritoneal surface and at the small bowel mesentery). Twenty-five patients (93%) had extensive tumors involving the small intestine, colon, sigmoid and/or rectum. Extensive liver, spleen or stomach involvement was seen in 11 (41%) patients. Five (19%) patients were diagnosed with frozen pelvis. Three patients (10%) were reported with tumor involvement at the renal vein, inferior vena cava or truncus coeliacus. 

### 3.2. Postoperative Treatment

Table 3 reported the postoperative treatments. There were four of 27 patients (15%) who did not continue chemotherapy. One patient had a poor performance status after surgery. She was ineligible for chemotherapy and died three months after surgery. The patient with low-grade serous adenocarcinoma received maintenance letrozole without continuing chemotherapy (died after 32 months). Two patients declined further chemotherapy after surgery: one patient died 15 months after surgery, and the other started paclitaxel/carboplatin chemotherapy at further progression 8 months after surgery but stopped after two cycles because of the side effects and died 17 months after surgery.

Most patients (n = 22, 81%) received postoperative chemotherapy with paclitaxel/carboplatin; in one patient, this was combined with bevacizumab. One patient received postoperative chemotherapy with cyclophosphamide/bevacizumab. Among the 23 patients who received further chemotherapy, five died within six months after surgery. Three patients had progressive disease within six months after surgery, and all of them received second-line chemotherapy.

Within 6–12 months after surgery, 11 patients (37%) had progressive disease. Platinum-based doublets were administered to six patients with the platinum-sensitive disease, of whom one received maintenance niraparib. Two patients received pegylated liposomal doxorubicin (PLD), two patients received paclitaxel weekly with bevacizumab, and one patient received gemcitabine for platinum-resistant disease.

In four patients with progressive disease > 12 months after first-line treatment, four different second-line regimens were given, i.e., single agent carboplatin, cyclophosphamide/bevacizumab, paclitaxel/carboplatin/bevacizumab and PLD/carboplatin followed by maintenance therapy the PARP inhibitor olaparib.

As part of the second-line treatment, six patients were treated with maintenance treatments after chemotherapy: three with bevacizumab and three with PARP inhibitors. One patient started with letrozole (Table 3).

### 3.3. Overall Survival

The median overall survival after surgery was 16 (IQR 5–21) months (95%CI 14–18) (Figure 2). At 24 months of follow-up, four patients (15%) were alive with the disease.

## 4. Discussion

The purpose of this study was to establish a detailed description of the treatment of individual patients and a report on overall survival. In this post hoc analysis of a large RCT, 27 patients with AEOC were included with the unresectable disease during abdominal exploration. We described in detail the subsequent treatments in our patients. Treatment after attempted surgery was divers, ranging from the cessation of treatment to, predominantly, one or several lines of chemotherapy with or without maintenance treatment with bevacizumab and/or PARP inhibitors. 

In our study, the median overall survival after surgery was 16 months (IQR 5–21). It is remarkable that 15% of the patients are still alive after two years. This is quite comparable with the study of Kaban et al., who reported a median OS of 22 months from the first treatment to death in patients with suboptimal cytoreduction [13], and Bland et al., who reported a median OS of 23 months for AEOC patients with suboptimal interval cytoreduction [14]. Both studies calculated the OS from the day chemotherapy started, while we calculated OS from the day of surgery.

### 4.1. CA-125

Every patient underwent interval CRS. The median CA-125 after NACT was 81 kU/L. Although previous studies demonstrated a correlation between the percentage of decrease in CA-125 after NACT and surgical outcome, we neither found an association between the preoperative value of CA-125 nor the reduction rate of CA-125 level after NACT and unresectable disease. One study reported a significant relation between CA-125 level and complete CRS in multivariable analysis, while all other studies did not perform a multivariable analysis [15,16,17]. Although the study by Gupta et al. found a significant correlation between a >95% decrease in preoperative CA-125 level and complete CRS, our study showed that this does not guarantee complete CRS. In our study, seven patients (26%) had a decrease of ≥95% [16]. 

### 4.2. Overall Survival

It must be noted that the survival of patients with unresectable disease is much worse than in patients in which a complete or optimal CRS is possible [5]. There is a paucity of data on survival outcomes comparing delayed CRS with no surgery (neoadjuvant chemotherapy only). To fill this knowledge gap, the GO SOAR2 study was designed [18].

In the PlaComOv study, both mandatory criteria of a proper selection of patients for CRS (via preoperative imaging) and a skilled surgical team to achieve complete cytoreduction were met. However, our results showed that in 8% of the cases, it was impossible to perform CRS due to extensive tumor lesions at the bowel and mesentery [8]. Unfortunately, from other studies, nothing is known about the number of patients with unresectable diseases. Only the study by Fagotti et al. described two out of 171 patients (1.1%) as unresectable due to retroperitoneal disease [19]. We believe that most studies included patients with unresectable diseases in the same group as those with suboptimal CRS. 

At present, there are no clear international treatment recommendations concerning further postoperative treatment strategies in the case of unresectable ovarian cancer [20]. In current practice, physicians discuss with their patients whether to continue their chemotherapy based on previous responses and toxicity. We described in detail the subsequent treatments in our patients. Treatment after attempted surgery was diverse. Most patients continued paclitaxel/carboplatin as part of the first-line treatment. The second-line treatment ranged from chemotherapy with or without maintenance treatment with bevacizumab and/or PARP inhibitors. Due to the small number and diverse treatments, we cannot comment on the treatment decisions or the best options. However, a number of patients who received second-line treatment and maintenance therapy survived for more than two years in our study. Therefore, maintenance therapy might be valuable for selected patients. 

### 4.3. Strength and Limitations

This study is a post hoc analysis in a prospective cohort of data from a multicenter randomized controlled trial. Patient characteristics and intra-operative and postoperative information were uniformly collected in an electronic database management platform which reduced the risk of missing data. Before any treatment was administered, a multidisciplinary tumor board was convened. All patients were centralized to the registered cancer hospitals and underwent CRS by experienced gynecologic oncologists, which ensured the maximal effort of the surgery. Despite some data limitations and bias due to the study’s nature, this study shed light on a subgroup of patients who are often overlooked.

## 5. Conclusions

This study indicates that patients with unresectable AEOC have poor survival rates. Currently, optimal treatment strategies in terms of survival benefits are still ill-defined. Further study of this specific group of patients is warranted. In order to provide a recommendation for these women, we advocate an (inter)national registry with a biobank of patients with unresectable cancer and comprehensive follow-up. Subsequently, an international multidisciplinary group of experts should write a clinical guideline with their treatment advice and evaluate this through prospective data collection of comparable patients. 

## Figures and Tables

**Figure 1 cancers-15-00072-f001:**
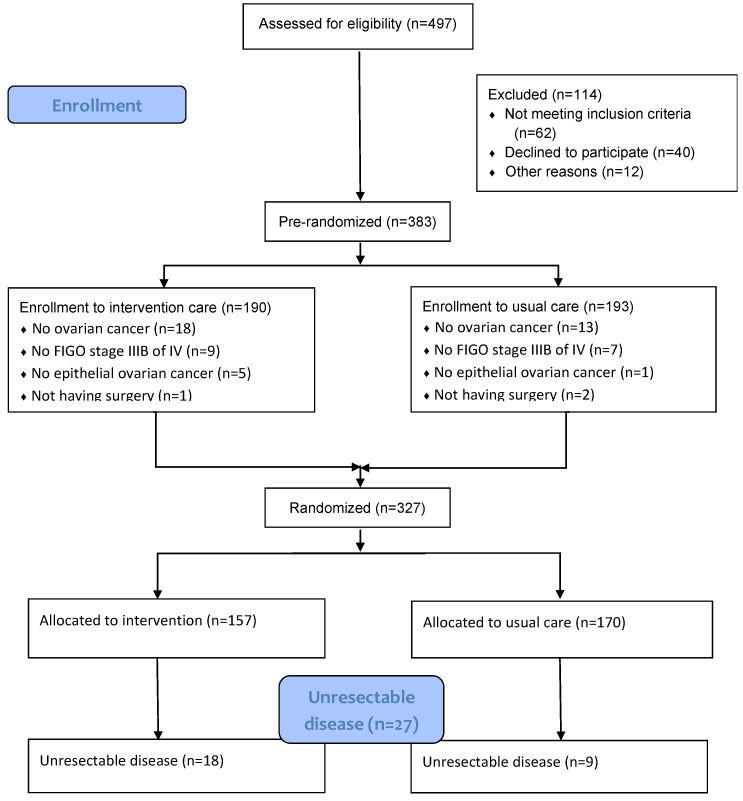
Consort flow diagram PlaComOv study.

**Figure 2 cancers-15-00072-f002:**
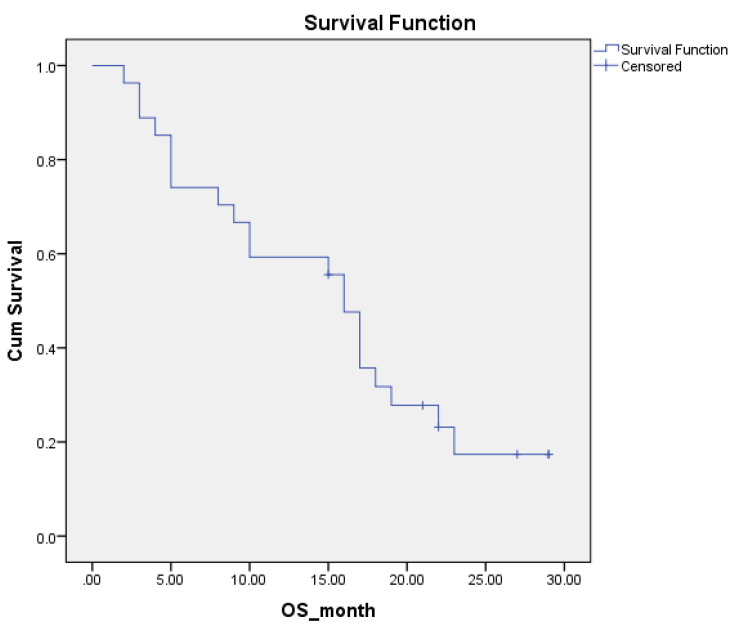
Kaplan–Meier curve of overall survival (n = 27).

**Table 1 cancers-15-00072-t001:** Patient characteristics.

Patient	Age (Year)	BMI (kg/m^2^)	FIGO Stage	CA-125 (Diagnosis) (kU/L)	CA-125 (NACT) (kU/L)	WHO ^1^	Dose Modification in NACT	Co-Morbidity ^2^	Polypharmarcy ^3^
1	67	47.9	IIIC	1715	98	2	No	−	+
2	65	20.7	IIIC	1681	599	2	No	−	+
3	72	24.5	IIIC	16,054	700 *	0	No	−	−
4	60	21.8	IIIC	586	41	2	No	−	−
5	62	31.6	IIIC	581	377	1	No	−	−
6	81	22.5	IIIC	2703	180	1	No	+	+
7	80	17.2	IIIC	5198	1873	1	No	−	−
8	73	31.6	IIIC	1300	79 *	1	No	−	−
9	76	22.9	IV	869	34 *	0	No	+	−
10	64	21	IV	2688	37 *	0	No	−	−
11	64	23.1	IIIC	220	98	1	No	−	−
12	76	19.8	IIIC	760	170	-	No	−	−
13	77	22	IV	130	81	0	No	−	−
14	75	19.5	IV	241	146	1	Yes	+	+
15	78	20.4	IV	1331	51 *	0	No	−	−
16	76	18.8	IV	11,239	3695	2	No	+	−
17	74	22.3	IV	2532	40 *	1	No	+	−
18	69	20.1	IV	237	27	1	No	+	−
19	61	22.5	IIIC	11,000	220 *	0	No	+	−
20	76	30.7	IV	290	64	0	Yes	−	−
21	78	22.3	IIIB	526	13.3 *	1	No	−	−
22	71	26.7	IV	550	38	0	No	−	−
23 †	28	20	IIIC	120	74.9	1	No	−	−
24	58	29.4	IIIC	470	270	0	No	−	+
25	72	27.7	IIIC	310	250	1	No	+	+
26	68	33.6	IV	660	51	1	No	−	−
27	68	28.3	IIIC	649	77	1	No	−	−

† low-grade serous cancer; * = ≥95% decrease; + = yes; − = no; NACT = neoadjuvant chemotherapy; ^1^ WHO = performance status (see method); ^2^ Comorbidity: + = 3 or more systems; − = 0–2 systems; ^3^ Polypharmacy: Five or more medicines for at least 90 days.

**Table 2 cancers-15-00072-t002:** Description of surgical findings.

Patient	Description of Surgical Findings
1	extensive peritoneal carcinomatosis, extensive tumor lesions entire bowel, mesentery and liver
2	extensive peritoneal carcinomatosis, extensive tumor lesions entire bowel and mesentery
3	extensive peritoneal carcinomatosis, tumor lesions up to 10 cm entire bowel, bladder, liver, spleen, diaphragm. Involvement renal vein by enlarged para aortic lymph nodes.
4	extensive peritoneal carcinomatosis, tumor lesions entire colon and small bowel, mesentery, liver, diaphragm, spleen, truncus coeliacus
5	extensive peritoneal carcinomatosis, extensive tumor lesions entire bowel and mesentery, no access to pelvis after adhesiolysis
6	extensive peritoneal carcinomatosis, extensive tumor lesions entire bowel, mesentery and liver
7	extensive peritoneal carcinomatosis, extensive tumor lesions entire bowel and mesentery, no access to pelvis after adhesiolysis
8	extensive peritoneal carcinomatosis, extensive tumor lesions entire bowel and mesentery
9	extensive peritoneal carcinomatosis, extensive tumor lesions entire bowel and mesentery, extensive tumor lesions in liver and spleen
10	extensive peritoneal carcinomatosis, extensive tumor lesions entire bowel, mesentery and mesocolon
11	extensive peritoneal carcinomatosis, extensive tumor lesions entire bowel and mesentery
12	extensive peritoneal carcinomatosis, extensive tumor lesions entire bowel and mesentery
13	extensive peritoneal carcinomatosis, extensive tumor lesions entire bowel and mesentery
14	extensive peritoneal carcinomatosis, extensive tumor lesions entire bowel and mesentery
15	extensive peritoneal carcinomatosis, extensive tumor lesions entire bowel and mesentery, all organs and block by adhesions
16	extensive peritoneal carcinomatosis, extensive tumor lesions entire bowel, mesentery and mesocolon
17	extensive peritoneal carcinomatosis, extensive tumor lesions entire bowel, mesentery, mesocolon and stomach
18	extensive peritoneal carcinomatosis, extensive tumor lesions entire bowel and mesentery
19 *	extensive peritoneal carcinomatosis, extensive tumor lesions entire bowel and mesentery
20	extensive peritoneal carcinomatosis, extensive tumor lesions entire bowel, mesentery, liver and stomach
21	extensive peritoneal carcinomatosis, extensive tumor lesions entire bowel, mesentery and liver, no access to pelvis after adhesiolysis
22	extensive peritoneal carcinomatosis, extensive tumor lesions entire bowel and mesentery, extensive tumor lesions in spleen
23 †	extensive peritoneal carcinomatosis, extensive tumor lesions in colon and mesocolon, spleen, pancreas, vessels liver, vena cava inferior
24	extensive peritoneal carcinomatosis, extensive tumor lesions entire bowel and mesentery
25	extensive peritoneal carcinomatosis, extensive tumor lesions entire bowel, mesentery and liver
26	extensive tumor in peritoneum, mesentery and liver
27 *	extensive peritoneal carcinomatosis, extensive tumor lesions entire bowel and mesentery, all organs and block by adhesions, mass in mesentery extending to the superior mesenterial artery

† low-grade serous cancer; * a diagnostic laparoscopy was performed prior to interval cytoreductive surgery.

**Table 3 cancers-15-00072-t003:** Postoperative treatment and overall survival.

Patient	Postoperative Treatment as Part of First-Line Treatment	Progression-Free Survival (Months)	2nd and Subsequent Treatment Lines	Overall Survival (Months)
1	-		-	2.9
2	TC		-	3.8
3	TC		-	3.9
4	TC		-	4.5
5	TC		-	5.1
6	TC		-	5.4
7	TC		-	5.7
8	TC maint bev	6	TC q3w (1 cycle)	8.1
9	TC	3	Wee1 kinase inhibitor and carboplatin (8 cycles)	9.3
10	TC	7	TC q3w	10.2
11	TC	4	PLD q4w- bev	10.3
12	-	3	-	15.3
13	TC	7 15	TC q3w TC (1 cycle)	16.2
14	TC	8	PLD q4w (1 cycle)	16.9
15	TC	9	TC, maint niraparib (3 weeks)	17.1
16	TC	7	paclitaxel weekly + bev, maint bev	17.7
17	-	8	TC weekly (2 cycles)	17.7
18	TC	8	paclitaxel weekly + bev, maint bev	18.1
19	TC	13	PLD/carboplatin q4w, maint olaparib	19.0
20	TC	10	PLD q4w	22.4
21	TC	9 20	TC weekly PLD q4w	22.9
22	cyclophosphamide/bev	19	carboplatin q3w (3 cycles)	23.1
23 †	Letrozole	18	continuing letrozole	31.5
24	TC	1	Letrozole	alive with disease >24 months
25	TC	13 28	Cyclofosfamide + bev, maint bev PLD/carboplatin q4w	alive with disease >36 months
26	TC	15	TC, maint olaparib	alive with disease >33 months
27	TC	8 17 21	gemcitabine PLD q4w TC q3w	alive with disease >28 months

† low-grade serous cancer; Q3w = every three weeks; TC = paclitaxel/carboplatin; PLD = pegylated liposomal doxorubicin; carb = carboplatin; bev = bevacizumab; maint = maintenance.

## Data Availability

Research data at Erasmus MC is generated, stored and made accessible in accordance with legal, academic and ethical requirements. This study, and all persons involved, have knowledge of and comply with the most recent version of the Erasmus MC Research Code, which complies with all current laws and regulations. Data will be handled by practicing the FAIR principles (Findable, Accessible, Interoperable and Reusable) according to the Handbook for Adequate Natural Data Stewardship (HANDS) developed by the Federation of Dutch UMCs. All research data is handled confidentially in accordance with legislation and conditions imposed by The Dutch Data Protection Authority. The research data from this study is stored in a long-term archive on secured network servers, with regular backup and limited access. In accordance with the Netherlands Code of Conduct for Scientific Practice, raw data is stored for a period of at least ten years. Permission for third persons to access the data will only be granted by the PI on certain conditions.

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
