# Peer review of "Unresectable Ovarian Cancer Requires a Structured Plan of Action: A Prospective Cohort Study"

_cancers, 2022, doi:10.3390/cancers15010072_

Round 1
Reviewer 1 Report
1. The literature of the study needs to be explained more elaborately. The introduction is very short to attract the readers. My suggestion is to describe the mechanism of poly adenosine diphosphate-ribose polymerase (PARP) inhibitors. Need to be expanded the information on Unresectable ovarian cancer in the Introduction section.
2. In the Methods Section, I would recommend using the Flow chart of the full research work that has been carried out.
3. In the Overall survival section, The authors mentioned that "The median overall survival after surgery was 16 (IQR 5-21) months (95%CI 14-18) 147 (Figure II)" but I don't even find Fig. II in the manuscript. Please explained it correctly.
4. Overall, this manuscript has novelty and after minor corrections, I would suggest proceeding.
Author Response
We appreciate your time for the thorough review and suggestions.
A1: We have expanded the introduction and added additional information regarding the available literature.
Despite extensive preoperative examinations, during abdominal exploration it may be found that cytoreductive surgery (CRS) is impossible because of extensive disease in patients with advanced stage epithelial ovarian cancer (AEOC). In those patients, sur-gery has to be abandoned because of unresectable disease. Patients with unresectable ovarian cancer are typically underreported or included in a suboptimal CRS (>1 cm residual tumor) group (1).
Ovarian cancer is the eighth most occurring cancer in women, with in 2020 almost 314,000 new cases and more than 207,000 deaths worldwide(2). At present, standard treatment of advanced-stage epithelial ovarian cancer (AEOC) consists of cytoreductive surgery (CRS) and platinum based chemotherapy (mainly 6 courses 3-weekly paclitaxel/carboplatin), followed by maintenance therapy with poly adenosine diphosphate-ribose polymerase (PARP) inhibitor in selected patients(3, 4). Timing of surgery may be as primary CRS or as interval CRS after 3 cycles of neoadjuvant chemotherapy(5). Complete resection of all macroscopic disease (at primary or interval surgery) is the strongest independent variable in predicting overall survival(6).
In case of unresectable disease for patients with AEOC, knowledge on further postoperative treatment and survival is limited.
To address this knowledge gap, we analyzed data of patients in whom surgery was abandoned due to unresectability after ab-dominal exploration(7, 8). The aim of this study was to establish a detailed account of individual patients’ treatment along with a report on overall survival.
We did not describe the mechanism of PARP inhibitors but referred to the references because we believe it distracts from the purpose of the study. The mechanism of the various chemotherapies is also not described.
Unfortunately, little is known about the group of patients with unresectable disease and what treatment they receive at later stages of the disease course. Only the study of Fagotti et al (SCORPION-NCT01461850) described two of 171 patients (1.1%) as unresectable due to retroperitoneal disease. We believe that most studies included the patients with unresectable disease in the same group as those with suboptimal CRS. Therefore, this manuscript draws attention to this issue.
Q2: In the Methods Section, I would recommend using the Flow chart of the full research work that has been carried out.
A2: Thank you for your suggestion to improve our manuscript. We have added the flow chart of full research work as presented in Figure I.
Q3: In the Overall survival section, The authors mentioned that "The median overall survival after surgery was 16 (IQR 5-21) months (95%CI 14-18) 147 (Figure II)" but I don't even find Fig. II in the manuscript. Please explained it correctly.
A3: We apologize, but this was a typo. We have updated the order of the figures.
C4: Overall, this manuscript has novelty and after minor corrections, I would suggest proceeding.
A4: Thank you very much for your constructive feedback. To our opinion, the above corrections contribute to the first draft.

Reviewer 2 Report
In the work Unresectable varian cancer requires a structured plan of action: a prospective cohort study, the authors consider the scheme of further treatment in cancer patients. Because ovarian cancer is one of the most serious problems of modern medicine, it is detected in late clinical stages and has a poor prognosis. Unfortunately, I have to draw the authors' attention to several aspects of the work that need improvement:
1. The introduction must be developed in order to emphasize the importance of the discussed research problem. In my opinion, it is worth presenting statistics on the diagnosis, treatment and quality of life of patients with ovarian cancer. The most common treatment regimens for ovarian cancer and measures of response to treatment should also be presented.
2. The description of the methodology should include the average age of the treated patients, the number of patients with individual histological types of ovarian cancer, describe the differences in CA125 concentrations, etc. The group of patients should be divided into subgroups in which patients with a similar diagnosis and in a comparable clinical condition would be gathered. What were the inclusion/exclusion criteria for patients? How could their age, body weight, presence of concomitant diseases influence the results of the study?
3. The results show that the patients were covered by very different treatment regimens - how can their survival results be comparable? Please expand the abbreviations of the drugs you use.
4. The discussion does not compare the obtained results with the results previously described in the literature. Please discuss how the patients' age, general health condition, time of diagnosis and treatment could have influenced the results. These differences in overall survival must be due to something.
I believe that the work raises a very important topic and can introduce changes in the treatment regimen of patients with ovarian cancer, but it requires a lot of refinement
Author Response
Thank you very much for your comprehensive review of our manuscript.
A1: We have updated the introduction and incorporated the comments mentioned above.
Despite extensive preoperative examinations, during abdominal exploration it may be found that cytoreductive surgery (CRS) is impossible because of extensive disease in patients with advanced stage epithelial ovarian cancer (AEOC). In those patients, surgery has to be abandoned because of unresectable disease. Patients with unresectable ovarian cancer are typically underreported or included in a suboptimal CRS (>1 cm residual tumor) group (1).
Ovarian cancer is the eighth most occurring cancer in women, with in 2020 almost 314,000 new cases and more than 207,000 deaths worldwide(2). At present, standard treatment of advanced-stage epithelial ovarian cancer (AEOC) consists of cytoreductive surgery (CRS) and platinum based chemotherapy (mainly 6 courses 3-weekly paclitaxel/carboplatin), followed by maintenance therapy with poly adenosine diphosphate-ribose polymerase (PARP) inhibitor in selected patients(3, 4). Timing of surgery may be as primary CRS or as interval CRS after 3 cycles of neoadjuvant chemotherapy(5). Complete resection of all macroscopic disease (at primary or interval surgery) is the strongest independent variable in predicting overall survival(6).
In case of unresectable disease for patients with AEOC, knowledge on further postoperative treatment and survival is limited.
To address this knowledge gap, we analyzed data of patients in whom surgery was abandoned due to unresectability after abdominal exploration(7, 8). The aim of this study was to establish a detailed account of individual patients’ treatment along with a report on overall survival.
Q2: The description of the methodology should include the average age of the treated patients, the number of patients with individual histological types of ovarian cancer, describe the differences in CA125 concentrations, etc. The group of patients should be divided into subgroups in which patients with a similar diagnosis and in a comparable clinical condition would be gathered. What were the inclusion/exclusion criteria for patients? How could their age, body weight, presence of concomitant diseases influence the results of the study?
A2: In the method section, we described all information that we had before we started the study. We added more information about the treated patients, the study design, measurements and analysis. In the results section, we described the results as average age of the treated patients etc. Because of the aim of the study (to address a knowledge gap of treatment for patients with unresectable disease, to establish detailed account of individual patients’ treatment along with a report on overall survival) and the small number of patients we could not divided patients into subgroups.
As you mentioned, we put more information about differences in CA125 concentrations etc. in the results.
Q3: The results show that the patients were covered by very different treatment regimens - how can their survival results be comparable? Please expand the abbreviations of the drugs you use.
A3: This is exactly why we would like to draw attention to this group of patients with unresectable disease. We don't know which treatment is really useful. And in our opinion, it is really a knowledge gap. It appears that a part of the patients continue the initiated therapy (carbo/taxol) but that a large part of the patients show progression of disease within 6 months. Since no treatment recommendation can be found in an international protocol, it appears that all patients received a different treatment. The purpose of this study is not to compare survival outcomes, but rather to describe these outcomes.
The abbreviations have been added.
Q4: The discussion does not compare the obtained results with the results previously described in the literature. Please discuss how the patients' age, general health condition, time of diagnosis and treatment could have influenced the results. These differences in overall survival must be due to something.
A4: We have changed the discussion section as much as possible.
Since no studies on unresectable ovarian cancer are published, we are not able to compare our results. Only the study of Fagotti et al (SCORPION-NCT01461850) described two of 171 patients (1.1%) as unresectable due to retroperitoneal disease. We believe that most studies included the patients with unresectable disease in the same group as those with suboptimal CRS.
Because of the small number of patients, it is not possible to make a statement as to why some patients live longer than others. In addition to the small number of patients, treatment is varied and therefore we cannot comment on treatment decisions or best options. This is described in the discussion as a possible explanation: 'the patients in our study who received second-line treatment and maintenance therapy survived more than two years. Therefore, maintenance therapy could be valuable for selected patients.'

Reviewer 3 Report
the Authors describe the results of a post-hoc analysis of the PlaComOv-study. Although this analysis does have merit, given the low number of patients enrolled I feel that it should be represented as a case series (the Results are somehow already presented in this fashion). I suggest the Authors to rework the paper and the conclusion accordingly.
Furthermore, the fact that this is a post-hoc analysis of the PlaComOv-study should be made clear from the abstract.
Author Response
A1: First of all, we would like to thank the reviewer for the efforts to review our manuscript. Thank you so much for your suggestion above.
With all authors, we have discussed the study design of this study as a case series according to your suggestion. In a case series you often describe cases that come under your care by chance. This is a post-hoc analysis of patients who participated in a large RCT. Therefore, in our opinion it is an observational cohort. For now, we did not change the study design. When you don’t agree with this, we can easily change this manuscript in a case series.
Q2: Furthermore, the fact that this is a post-hoc analysis of the PlaComOv-study should be made clear from the abstract.
A2: In the abstract, we have mentioned that this study was a post hoc analysis of the PlaComOv study.

Round 2
Reviewer 2 Report
I would like to thank the Authors for taking my comments into account in their work. The introduction and description of the study group have been supplemented, the discussion has also been developed, and additional literature sources have been quoted.
However, the main problem of the work remains - as emphasized by the authors themselves - the small number of patients subjected to the study. It is dangerous to draw conclusions based on such an unrepresentative group. I would still suggest diversifying the study group in terms of age, BMI, CA 125 concentration and so on - since the Authors themselves admit that "due to the small number of patients we could not divided patients into subgroups" one should consider whether to refrain from publishing until enough results are collected. Nevertheless, I am submitting the article for publication, with the emphasis that it should rather be treated as preliminary research, which will be continued.
Reviewer 3 Report
thanks for having considered my comments. I cannot say I agree with the decision to present this analysis as a cohort. However, since my "collagues reviewers" have not raised this issue, I may be wrong, for sure. I leave any further comment to the Editor, and wish all the best to the Authors for their publication.